# Whole-Transcriptome Sequencing Analyses of Nuclear Antixoxidant-1 in Endothelial Cells: Role in Inflammation and Atherosclerosis

**DOI:** 10.3390/cells11182919

**Published:** 2022-09-18

**Authors:** Varadarajan Sudhahar, Yang Shi, Jack H. Kaplan, Masuko Ushio-Fukai, Tohru Fukai

**Affiliations:** 1Vascular Biology Center, Medical College of Georgia, Augusta University, Augusta, GA 30912, USA; 2Department of Pharmacology and Toxicology, Medical College of Georgia, Augusta University, Augusta, GA 30912, USA; 3Charlie Norwood Veterans Affairs Medical Center, Augusta, GA 30901, USA; 4Department of Population Health Science, Medical College of Georgia, Augusta University, Augusta, GA 30912, USA; 5Department of Biochemistry and Molecular Genetics, University of Illinois College of Medicine, Chicago, IL 60607, USA; 6Department of Medicine (Cardiology), Medical College of Georgia, Augusta University, Augusta, GA 30912, USA

**Keywords:** antioxidant-1, copper, transcriptome sequencing, ROS, inflammation, endothelial cells, atherosclerosis

## Abstract

Inflammation, oxidative stress, and copper (Cu) play an important role in cardiovascular disease, including atherosclerosis. We previously reported that cytosolic Cu chaperone antioxidant-1 (Atox1) translocates to the nucleus in response to inflammatory cytokines or exogenous Cu and that Atox1 is localized at the nucleus in the endothelium of inflamed atherosclerotic aorta. However, the roles of nuclear Atox1 and their function are poorly understood. Here we showed that Atox1 deficiency in ApoE^−/−^ mice with a Western diet exhibited a significant reduction of atherosclerotic lesion formation. In vitro, adenovirus-mediated overexpression of nuclear-targeted Atox1 (Ad-Atox1-NLS) in cultured human endothelial cells (ECs) increased monocyte adhesion and reactive oxygen species (ROS) production compared to control cells (Ad-null). To address the underlying mechanisms, we performed genome-wide mapping of Atox1-regulated targets in ECs, using an unbiased systemic approach integrating sequencing data. Combination of ChIP-Seq and RNA-Seq analyses in ECs transfected with Ad-Atox1-NLS or Ad-null identified 1387 differentially expressed genes (DEG). Motif enrichment assay and KEGG pathway enrichment analysis revealed that 248 differentially expressed genes, including inflammatory and angiogenic genes, were regulated by Atox1-NLS, which was then confirmed by real-time qPCR. Among these genes, functional analysis of inflammatory responses identified CD137, CSF1, and IL5RA as new nuclear Atox1-targeted inflammatory genes, while CD137 is also a key regulator of Atox1-NLS-induced ROS production. These findings uncover new nuclear Atox1 downstream targets involved in inflammation and ROS production and provide insights into the nuclear Atox1 as a potential therapeutic target for the treatment of inflammatory diseases such as atherosclerosis.

## 1. Introduction

Inflammation contributes to the initiation and progression of atherosclerotic lesion, which is associated with immunity [1], clonal hematopoiesis of indeterminate potential (CHIP) [2], and oxidative stress [3,4]. The causal role of inflammation in atherosclerosis and cardiovascular risk has been supported by recent clinical trials, such as the canakinumab Anti-inflammatory Thrombosis Outcomes Study (CANTOS) on patients with established atherosclerotic disease [5,6]. In addition, excessive reactive oxygen species (ROS)/oxidative stress also play an important role in vascular inflammatory disease, such as atherosclerosis, by inducing expression of redox-sensitive inflammatory genes such as VCAM-1 and ICAM-1 in endothelial cells (ECs), oxidative modification of lipoproteins [3], and macrophage activation [4]. Furthermore, copper (Cu), an essential micronutrient, has also been implicated in inflammation and atherosclerosis [7,8]. Implanting a Cu cuff promotes vascular neointima thickening [9], whereas Cu chelator treatment inhibits inflammation in vascular tissue and development of atherosclerotic lesion in ApoE^−/−^ mice [10], as well as neointimal formation in response to vascular injury [11,12]. Cu plays an important role in inflammatory responses involved in innate and adaptive immunity [13,14]. For example, Cu deficiency or Cu chelator decreased leukocyte adhesion to the activated ECs and adhesion molecule expression, such as ICAM-1/VCAM-1 [15]. Despite the critical role of inflammation, ROS, and Cu in atherosclerosis, the mechanistic link between Cu, ROS, and inflammation in atherosclerosis remains unknown.

Antioxidant-1 (Atox1) [8,16,17,18] is known to function as a cytosolic Cu chaperone with 68 amino acid residues and is highly conserved among mammalian species. Atox1 at the cytosol transfers Cu(I) to the Cu-transporting ATPase, ATP7A, at the trans-Golgi network (TGN) for delivery of Cu to the secretory pathway or secretory cuproenzymes such as extracellular SOD (ecSOD, SOD3) or lysyl oxidase to regulate their activity [8,18,19]. Importantly, Hamza et al. [20] initially reported that Atox1 is localized not only in the cytosol to function as a Cu chaperone for ATP7A, but also localized in the nucleus in human cancer cell line HeLa cells. This finding was further confirmed by us and others in other cell types [16,21,22,23,24,25,26,27,28,29,30]. For example, we demonstrated that: Atox1 is predominantly localized at the nucleus in the endothelium of inflamed atherosclerotic aorta compared to control aorta [17]; Atox1 translocates from the cytosol to the nucleus in response to exogenous Cu to function as a Cu-dependent transcription factor for cyclin D1 to promote cell proliferation [25]; G-protein-coupled receptor agonist angiotensin II stimulates Atox1 nuclear translocation to promote transcription of ecSOD in vascular smooth muscle cells, which protects against oxidative stress in hypertension [29]; inflammatory cytokine TNFα rapidly stimulates nuclear translocation of Atox1 in Cu- and TRAF4-dependent manner to upregulate cytosolic NADPH oxidase organizer, p47phox in ECs, which promotes ROS-dependent inflammatory angiogenesis in vivo [16,17,31]; and nuclear translocation of Atox1 potentiates activin A-induced cell migration and colony formation in colon cancer [27]. Consistent with our findings, others have also reported nuclear Atox1 in various cancers [21,22,32,33,34]. Jin et al. reported that ATOX1 binds to the promoter of mediator of DNA damage checkpoint 1, which enhances its transcriptional activity and tumor growth [33]. Chen et al. reported that APEX2-based proximity labeling of Atox1 identifies cysteine-rich protein 2 (CRIP2) as a nuclear Cu-binding protein that regulates autophagy activation in lung cancer H1299 cells [32]. Despite these critical advances in our knowledge supporting the transcription factor function of Atox1, a genome-wide analysis of Atox1 targeted genes in ECs and their function has not been described.

In this study, we performed the genome-wide mapping of Atox1-regulated target gene analysis, using unbiased systemic approaches such as chromatin immunoprecipitation followed by high-throughput sequencing (ChIP-Seq) and RNA-sequencing (RNA-Seq) in human ECs transfected with Atox1 with nuclear-targeted sequence (Atox1-NLS). Our broad analysis identified a number of differentially expressed genes (846 upregulated and 541 downregulated genes) regulated by nuclear Atox1. Moreover, motif enrichment assay and a KEGG pathway enrichment analysis indicated that Atox1-NLS increases the genes involved in inflammation and ROS production in ECs. These findings provide insights into nuclear Atox1 as a potential therapeutic target for the treatment of inflammatory diseases such as atherosclerosis.

## 2. Methods

### 2.1. Animals

Atox1^−/−^ mice were obtained from Mutant Mouse Regional Resource Centers and backcrossed eight times to C57Bl/6J (Jackson laboratory, Bar Harbor, ME, USA). ApoE^−/−^/Atox1^−/−^ mice were generated by crossing Atox1^−/−^ and ApoE^−/−^ mice (Jackson Laboratory, Bar Harbor, ME, USA). ApoE^−/−^/Atox1^−/−^ or ApoE^−/−^ mice were maintained on regular chow for 7 weeks and then on a Western diet (TD88137; ENVIGO, Indianapolis, IN, USA) for 16 weeks. Diet and water were provided ad libitum. The use of mice was in accordance with the National Institutes of Health Guide for the Care and Use of Laboratory Animals and relevant ethical regulations. Studies were carried out in accordance with guidelines approved by the Institutional Animal Care and Use Committee of Augusta University.

#### Atherosclerotic Lesion Analysis

After being fed a Western diet, mice were anesthetized and perfused with phosphate-buffered saline (PBS) and subsequently paraformaldehyde (4%) in PBS. Then, periadventitial fat and connective tissue were removed from the aortas. Aortas were stained with 2% Oil Red O (Sigma Aldrich, St. Louis, MO, USA) and opened longitudinally, and photographs were taken. The aortic atherosclerotic lesions were quantified using Image-Pro Plus software (Media Cybernetics, Rockville, MD, USA). Additionally, the upper one-third of the heart was excised and embedded in an optimum cutting temperature compound. Then, the tissue was cryosectioned (5 μm thick). Sections were stained with Oil Red O or hematoxylin to evaluate the size of necrotic cores or atherosclerotic lesions in the aortic sinus. Image-Pro Plus software (Media Cybernetics, Rockville, MD, USA) was used for quantification.

### 2.2. Cell Culture and Reagents

Human umbilical vein endothelial cells (HUVECs) (Lonza, Portsmouth, NH, USA) were cultured in a flask coated with 0.1% gelatin using EndoGRO basal medium (Millipore, Burlington, MA, USA) containing 5% fetal bovine serum (FBS) and used for experiments until Passage 6. Human aortic endothelial cells (HAECs) (Cell Application Inc. San Diego, CA, USA) were grown in human endothelial cell growth medium (Cell Application Inc.) containing 10% FBS and used for experiments until Passage 7. Proinflammatory cytokine cocktail containing TNF-α (10 ng/mL), IL-1β (10 ng/mL), and IL-6 (10 ng/mL) was used in in vitro experiments. siRNAs were transfected using Oligofectamine (Invitrogen, Waltham, MA, USA) in HUVECs for 3 h, followed by 48 h growth. The sequence for human CD137 siRNA is 5′-AAGCAGTTACTACAAGGATCC-3′; human CSF1 siRNA is 5′-GAUCCAGUGUGCUACCUUAAGAAGG-3′; human IL5RA siRNA is 5′-GCAAAGGAAUGUUAAUCUAGAAUAT-3′. Human Atox1 siRNA was purchased from Ambion (Austin, TX, USA). THP-1 monocyte cells were grown in RPMI medium with 10% FBS.

#### 2.2.1. siRNA and Adenovirus Transfection

HUVECs were incubated with either Ad-Null (control) or Ad. Atox1-NLS or Ad-catalase (University of Iowa Viral Vector Core, Iowa City, IA, USA) in 5% FBS containing culture medium for 24 h, followed by incubation in culture medium without virus for 24 h before experiments. For nuclear-targeted Atox1 (Flag-Atox1-NLS) constructs, a tripartite NLS sequence (PKKKRKVD) derived from the SV40 large T antigen was fused to the C terminus of Flag-tagged WT-Atox. For siRNA transfection, HUVECs were grown to 50% confluence and transfected with control siRNA, CD137 siRNA, IL5RA siRNA, CSF1 siRNA, or Atox1 siRNA (30 nM) using Oligofectamine. siRNAs were transfected using Oligofectamine (Invitrogen, Waltham, MA, USA) in HUVECs for 3 h, followed by 48 h growth. For double transfection of siRNA and Ad-virus, cells were transfected with siRNAs as described above. After 16 h of siRNA transfection, cells were transfected with Ad-null or Ad-Atox1-NLS or Ad-Catalase. Cells were used for experiments at 48 h after adenovirus transfection.

#### 2.2.2. Immunofluorescence Analysis

HUVECs were grown on glass coverslips. Cells were washed quickly in ice-cold PBS, fixed with paraformaldehyde (4%) in PBS for 10 min at room temperature, and permeabilized in Triton X-100 (0.05%) in PBS for 5 min. Then, cells were rinsed with PBS, quenching aldehydes with 50 μM NH_4_Cl, and washed with PBS for 10 min each. After incubation for 1 h in blocking buffer (PBS+ 3%BSA), cells were incubated with flag antibody (F7425, Sigma Aldrich, St. Louis, MO, USA) or Atox1 antibody [25] for 18 h at 4 °C. Then, cells were rinsed with PBS and incubated in Alexa Fluor 488-conjugated goat anti-rabbit IgG for 30 min at room temperature. Finally, cells on coverslips were mounted onto glass slides using Vectashield (Vector Laboratories, Newark, CA, USA) and photographed using confocal microscopy.

#### 2.2.3. Nuclear/Cytosolic Fractionation Assay and Immunoblotting

Nuclear/cytoplasmic fractionation from HUVECs was performed using the NEPER Nuclear and Cytoplasmic Extraction Reagents Kit (ThermoFisher, Waltham, MA, USA) according to the manufacturer’s protocol. For immunoblotting, cell lysates from nuclear/cytosolic fractionation or total cell lysate were separated using SDS-polyacrylamide gel electrophoresis and transferred to nitrocellulose membranes and blocked with blocking buffer (PBS containing 5% nonfat dry milk and 0.1% Tween 20) for 1 h. Then, the membrane was incubated overnight with primary antibodies (anti-Atox1 (this laboratory), anti-flag (Sigma, St. Louis, MO, USA), anti-actin (Santa Cruz, Dallas, TX, USA), GADPH, and P84 antibody (GeneTex, Irvine, CA, USA). After incubation with Goat Anti-Rabbit IgG-HRP Conjugate or Goat Anti-Mouse IgG-HRP Conjugate (Bio-Rad, Hercules, CA, USA), proteins were detected by ECL chemiluminescence.

#### 2.2.4. Monoclonal Atox1 Antibody Production

Mouse monoclonal Atox1 antibody (Z31112) was developed by using baculovirus particles displaying the surface glycoprotein gp64-fusion protein as the immunizing agent, as we previously described [16]. Briefly, the Atox1 cDNA was ligated into the gp64 gene to create a fusion protein that is expressed in the viral surface, as previously described [16]. Positive clones were selected by Western blotting using recombinant Atox1 protein or cells overexpressing Atox1 by adenovirus [25,26].

#### 2.2.5. Chromatin Immunoprecipitation (ChIP) and Sequencing

HUVECs grown to confluence were transfected with Atox1-NLS for 48 h. ChIP assays were performed by following the EpiTect ChIP One-Day Kit protocol (Qiagen, Germantown, MD, USA), as we described previously [16]. Briefly, cells were incubated with 1% formaldehyde and lysed with lysis buffer. Then, cells were sonicated and diluted with dilution buffer. The lysate was incubated with monoclonal Atox1 antibody (Z31112) (see above) or normal mouse IgG as control. ChIP-enriched DNA was subjected to library preparation using a ChIP-Seq DNA Sample Prep kit (Illumina, San Diego, CA, USA). Sequencing was performed using an Illumina HiSeq 2500 platform (GTAC, Washington University, St. Louis, MO, USA). For validation, the primer sequence was used as shown in Appendix A.

#### 2.2.6. ChIP Sequencing Analysis

The FASTQ ChIP-Seq read files were aligned to the reference human genome build hg19 using Bowtie 2 (v2.4.2, Johns Hopkins University, Baltimore, MD, USA) with default settings, and the generated BAM files were sorted and used as input files by MACS (v2.2.7.1, Harvard University, Cambridge, MA, USA) with default settings for peaks calling, where false discovery rate (FDR) <5% was used to call an Atox1 binding peak. ChIPseeker (v1.32.0, Southern Medical University, Guangzhou, China) was used for the annotations of those Atox1 binding peaks, and Integrative Genomics Viewer (v2.10.0, Broad Institute, Cambridge, MA, USA) was used for the visualization of the peaks at specific genomic regions. HOMER (v4.11, The University of California, San Diego, CA, USA) with default settings was used for motif analysis, where those significantly enriched motifs were identified using FDR < 5%.

#### 2.2.7. RNA Sequencing

Total RNA was extracted using the RNeasy kit (Qiagen, Germantown, MD, USA), and RNAs were quantified by spectrophotometry (ND-2000, NanoDrop, ThermoFisher, Waltham, MA, USA). The quality of RNAs was determined using electrophoresis (2100 Bioanalyzer, Agilent Technologies, Santa Clara, CA, USA) prior to next-generation sequencing in the Genome Technology Access Center (GTAC) at Washington University, St. Louis, MO, USA. Samples with RNA integrity number (RIN) values >8.0 were subjected to poly-A selection, chemical fragmentation, random hexamer priming, cDNA synthesis, and adapter-ligation using the TruSeq RNA Library Prep Kit (Illumina, San Diego, CA, USA), followed by paired-end multiplexed sequencing (HiSeq 2500, Illumina) according to the manufacturer’s instructions.

#### 2.2.8. RNA-Seq Analysis

The FASTQ RNA-Seq read files were quantified using Kallisto (v0.46.1, California Institute of Technology, Pasadena, CA, USA) with the transcriptome indices built on the cDNA FASTA files from human genome assembly GRCh38. Genes with lower than 10 total read counts were filtered for subsequent differential expression analysis. Differential expression analysis was performed using DESeq2 (v1.32.0, The University of North Carolina at Chapel Hill, Chapel Hill, NC, USA) by comparing Ad.Atox1-NLS and Ad.null samples, and genes with fold changes greater than 2.5 (>2.5 as upregulated and <0.4 as downregulated) and adjusted *p* < 0.05 were considered differentially expressed. The Gene Ontology (GO) Consortium, Molecular Signatures Database (MSigDB), and Kyoto Encyclopedia of Genes and Genomes (KEGG) pathway databases were used to perform GO and pathway enrichment analysis using R package clusterProfiler (v4.0.2, Southern Medical University, Guangzhou, China).

#### 2.2.9. Quantitative Real-Time PCR

Total RNA of HUVECs was isolated using TRI Reagent (Molecular Research Center Inc. Cincinnati, OH, USA). Reverse transcription was carried out using the high capacity cDNA reverse transcription kit (Applied Biosystems, Waltham, MA, USA) using 2 µg of total RNA. Quantitative PCR was performed with the SYBR Green PCR kit (Qiagen, Germantown, MD, USA) and ABI Prism 7000. The primer sequence for specific genes (Appendix A). Samples were run in duplicates to reduce variability. Expression of genes was normalized with 18S and expressed as fold change of relative mRNA expression compared to control siRNA or Ad-Null.

#### 2.2.10. Monocyte Adhesion Assay

HUVECs were transfected with Ad-Null or Ad-Atox1-NLS and grown in 24-well-plates. THP-1 cells were labeled with Cell Tracker Green CMFDA (Invitrogen, Waltham, MA, USA). The labeled THP-1 cells were added to the confluent EC monolayer. Cultures were incubated in a CO_2_ incubator for 1 h. Nonadherent cells were removed from the plate by gentle rinsing with PBS, and the number of adherent cells was determined by confocal microscopy. Cells were counted in five random 20× fields per well.

#### 2.2.11. ROS Measurement

ROS production in HUVEC cells was detected by incubating the cells with 20 μM of CM-H2DCFDA (5-(and-6)-chloromethyl-2′,7′-dichlorodihydrofluorescein diacetate, acetyl ester, Invitrogen, Waltham, MA, USA) for 6 min and photographed by confocal microscopy using the same exposure condition in each condition. Relative DCF-DA fluorescence intensity was analyzed using LSM software as we reported previously [16].

### 2.3. Statistical Analysis

All experiments were performed at least three times, and all values were expressed as mean ± SEM. Comparison between groups was analyzed by unpaired Student’s two-tailed *t*-test for two groups or one-way analysis of variance (ANOVA), followed by Bonferroni’s post hoc analysis for experiments with more than two subgroups. Statistical tests were performed using Prism v9 (GraphPad Software, San Diego, CA, USA).

## 3. Results

### 3.1. Role of Atox1 in Inflammatory Disease

We initially confirmed our previous observation [16,17] that Atox1 is mainly localized in the cytosol in unstimulated, basal cultured human umbilical vein endothelial cells (HUVECs), or human aortic endothelial cells (HAECs) and that Atox1 is translocated to the nucleus in response to the proinflammatory cytokines (tumor necrosis factor (TNF)-α, interleukin (IL)-1β, and IL-6), which are involved in vascular pathology (Figure 1A, Appendix A) [35,36]. This is consistent with our previous report that Atox1 is localized at the nucleus in the endothelium of inflamed atherosclerotic aorta [17]. To investigate the functional significance of Atox1 in atherosclerosis in vivo, we used either Atox1^−/−^ mice crossed with ApoE^−/−^ mice (Atox1^−/−^/ApoE^−/−^) or ApoE^−/−^ (control) mice on a 4-month Western diet. Oil Red O staining revealed a significant decrease in atherosclerotic lesions in the surface lesion area (Figure 1B) and the aortic sinus (Figure 1C) in Atox1^−/−^/ApoE^−/−^ mice, as compared to control mice. Masson Trichrome staining of collagen in the aortic sinus showed significantly higher fibrosis in ApoE^−/−^/Atox1^−/−^ mice as compared to control mice (Figure 1C). These results suggest that Atox1 localized in the nucleus plays an important role in inflammatory disease, such as atherosclerosis.

### 3.2. Nuclear Atox1 Increases Inflammatory Responses and ROS Production in ECs

To determine the roles of nuclear Atox1 in inflammatory responses in cultured ECs, we generated adenovirus expressing nuclear-targeted Atox1 (Ad.Atox1-NLS) in which a tripartite nuclear localization sequence (NLS, PKKKRKVD) derived from the SV40 large T antigen was fused to the C terminus of Flag-tagged Atox1, as we previously reported [25]. Immunofluorescence and immunoblotting analyses confirmed that Atox1-NLS was predominantly expressed in the nucleus in ECs transduced with Ad.Atox1-NLS as compared to Ad.null (control) (Figure 2A,B, Appendix A). We also found that Atox1 expression is increased in total lysate from Ad.Atox1-NLS transfected ECs (Appendix A). Functional significance of nuclear Atox1 showed that Ad.Atox1-NLS significantly increased monocyte adhesion (Figure 2C) and ROS production, as measured by DCF fluorescence (Figure 2D), which were inhibited by catalase overexpression (Figure 2C,D) in ECs. These results suggest that nuclear Atox1 overexpression is sufficient to induce inflammatory responses and ROS production in ECs.

### 3.3. Atox1 Gene Binding and Gene Expression Signatures

To characterize the genes differentially expressed in response to overexpression of nuclear Atox1, we performed genome-wide analysis by combining ChIP-Seq and RNA-Seq data sets in HUVECs treated with Ad.Atox1-NLS or Ad.null, as outlined in Figure 3A. HUVECs were transfected with Ad-Atox1 NLS or Ad.null for 48 h, followed by RNA isolation, and subjected to cDNA library preparation and deep sequencing. Figure 3A shows the summary of the differentially expressed Atox1-bound genes, i.e., genes that are identified as differentially expressed in RNA-Seq and have at least one Atox1 binding peak identified in ChIP-Seq. We find that Atox1 binds to 541 (85%) among 638 downregulated genes (Appendix A; designated as Atox1 downregulated targets) and 846 (85%) among the 997 upregulated genes by Atox1 in RNA-Seq (Appendix A; designated as Atox1 upregulated targets). Figure 3B shows the expression levels for those 541 downregulated and 846 upregulated genes, and Figure 3C shows the pattern for all genes analyzed in RNA-Seq with red dots representing those differentially expressed genes (DEGs).

Using MACS, a total of 46,075 Atox1 binding peaks were called at an FDR of 5%. Figure 3D shows the intensities of those peaks at the transcription start sites (TSSs), and Figure 3E shows the genomic distribution of those peaks relative to functional categories across the genome, where the majority of Atox1-binding peaks were localized in introns (37%) and intergenic regions (54%) (Figure 3E). However, after normalization to the size of these functional regions, it became clear that Atox1-binding peaks were enriched in gene promoters and 5′-UTRs (Figure 3E,F). Motif enrichment analysis revealed ten highly enriched motifs, including the Atox1 binding motif (Atox1-RE: GAAAGA) identified in our previous work [25] and other motif families (transcription factor, PRDM14, TEC1, bHLH80) (Figure 3G,H). These results indicate that nuclear Atox1 regulates gene expression via Atox1-RE or other motif families (i.e., coactivator).

### 3.4. Analysis of Nuclear Atox1-Regulated Genes

Further genomic annotations of those Atox1 binding peaks identified in ChIP-Seq revealed that a total of 20,861 genes contained at least one peak within 5 kb upstream or downstream of their genomic loci, and among them 541 genes are downregulated and 846 are upregulated by Atox1 in RNA-Seq (Figure 3A and Figure 4A), which results in 1387 DEGs. Next, we performed the pathway and Gene Ontology (GO) enrichment analysis using those 1387 Atox1-regulated DEGs (Figure 4A). Specifically, we conducted enrichment analysis using the following databases: GO Consortium (including GO terms in biological process, BP, molecular function, MF, and cellular components, CC), Molecular Signatures Database (MSigDB), and Kyoto Encyclopedia of Genes and Genomes (KEGG) pathways. GO analysis demonstrates that 1101 BP GO terms, 90 CC GO terms, and 136 MF GO terms were significantly enriched (Appendix A). In KEGG pathway analysis, those 1387 DEGs were classified into 305 biological pathways, and especially, and 28 pathways were significantly enriched (Figure 4A, Appendix A). Particularly, these DEGs were significantly enriched in the inflammatory TNF signaling pathways, cell cycle, p53 signaling, and Hippo signaling pathways (Figure 4B, Appendix A). Based on the expression of the DEGs in those four pathways, we identified several candidate Atox1 regulated genes involved in those pathways and further validated their differential expression levels between Ad.Atox1-NLS and Ad.null by qRT-PCR, which are listed as follows: inflammatory TNF signaling pathways—TRAF1, CSF1, MAP3K14 (Figure 4C); cell cycle pathways—CDC25A, CREEBP, CDC45; P53 signaling pathways—SENS2, MDM2, GTSE1; Hippo signaling pathways—TP73, TAZ, FZD5 (Appendix A). These analyses suggest that Atox1 regulates inflammation and other signaling pathways.

### 3.5. Genome-Wide Identification of Direct Atox1 Target Genes

We next analyzed the expression changes of nuclear Atox1 directly targeted genes, i.e., genes differentially expressed in response to Atox1-NLS in RNA-Seq and having at least one Atox1 binding peak in ChIP-Seq and at least one Atox1 binding motif sequence (GAAAGA) within 3 kb of their promoters, and this analysis identified 248 such genes, where 154 genes were upregulated and 94 genes were downregulated in HUVEC overexpressing Atox1-NLS compared to control (Figure 5A, Appendix A), and their expression pattern was shown in Figure 5B. We further performed pathway enrichment analysis based on those genes using the KEGG pathway database, and those significantly enriched pathways are shown in Figure 5C. Among them, we validated the expression of two groups of genes that are related to specific biological processes using qRT-PCR, which are inflammation-related genes, including ADRB1, CSF1, ERBB3, DUSP16, TNFSF11, MAFA, CD137, and IL5RA (Figure 5D, Appendix A) and angiogenic-related genes, including CCN2, LMO2, ADIPOQ and PIM1 (Appendix A). Furthermore, in both groups, the Atox1 binding peaks near the transcription start site (TSS) of several selected Atox1-regulated targeted genes are shown, which include ADRB1, CSF1, IL5RA, and CD137 (inflammation-related genes, Figure 6A) and CCN2, PIM1, LMO2 and MDM2 (Appendix A). Furthermore, these Atox1-regulated targets were further validated by ChIP-PCR, showing the enrichment of Atox1-bound chromatin peaks (Figure 6B and Appendix A).

### 3.6. CD137, IL5RA, and CSF1 as New Atox1 Targets to Regulate Inflammation and ROS Production in ECs

Both RNA-Seq and ChIP-Seq analysis identified CD137 (TNFRSF9), IL5RA, and CSF1 as new direct targets of Atox1 which regulate inflammatory responses. To determine the functional significance of Atox1-NLS-induced CD137, IL5RA, and CSF1 expression, we examined their roles in monocyte adhesion on ECs in vitro using THP-1 monocytes and HUVECs. CD137, IL5RA, and CSF1 siRNA treatment (Figure 7A) significantly decreased Atox1-NLS-induced monocyte adhesion in HUVECs (Figure 7B). Furthermore, CD137 siRNA, but not either IL5RA or CSF siRNA, significantly decreased Atox1-NLS-induced ROS production in ECs, as measured by DCF fluorescence (Figure 7C). We next examined the roles of endogenous Atox1 in proinflammatory cytokine induced CD137, IL5RA, or CSF1 mRNA expression. Atox1 siRNA significantly decreased CD137 and IL5RA mRNAs, but not CSF1 mRNA, in response to proinflammatory cytokines (Appendix A).

## 4. Discussion

Antioxidant-1 (Atox1) in the cytosol functions as a Cu chaperone to deliver Cu to ATP7A in the secretory pathway and is essential for Cu homeostasis. In addition, we and others showed that Atox1 is translocated from the cytosol to the nucleus to stimulate transcription of several genes regulating cell growth and ROS-dependent inflammatory gene expression in ECs in response to Cu or the inflammatory cytokine, TNFα [16,17,31]. However, downstream targets and the functional significance of nuclear Atox1 are still largely unknown. Here we performed whole-transcriptome sequencing analyses of ECs transfected with Atox1-NLS to identify genes regulated by nuclear Atox1. Using unbiased systemic approaches by combining RNA-Seq with ChIP-Seq, 248 DEGs were seen to be directly regulated by nuclear Atox1 via Atox1-RE, while 1139 DEGs were mediated via DNA binding motifs other than Atox1-RE. A KEGG pathway enrichment analysis revealed that nuclear Atox1 regulates genes related to inflammatory responses, including CD137, IL5RA, and CSF1, confirmed by real-time qRT-PCR and functional assays. Functionally, Atox1-deficient mice showed decreased atherosclerosis development on a high-fat diet. These findings uncover new nuclear Atox1 downstream target genes involved in inflammation and highlight the potential for targeting nuclear Atox1 as a treatment for inflammatory diseases.

This study performed genome-wide mapping of Atox1 target genes in ECs by combining ChIP-seq with RNA-Seq for the first time and identified the 46,075 genome-wide Atox1 binding peaks (annotated to 20,861 genes) in ECs overexpressing nuclear-targeted Atox1 compared to control. Among the genes around these peaks, 846 genes were upregulated, while 541 genes were downregulated by Atox1-NLS. The majority of Atox1 binding sites were in the intergenic regions and introns, but most of the Atox1 binding peaks were enriched proximal to the TSS, suggesting transcriptional regulatory functions for Atox1. An unbiased genome-wide motif search showed Atox1 binding to the DNA consensus sequence (Atox1-RE, GAAAGA) in the promoters, consistent with our previous reports [25]. Furthermore, PRDM4, TEC1, and bHLH80 motifs were enriched at Atox1-bound sites, suggesting that Atox1 binds to DNA via these transcription factors. Together with our current findings, which show other motifs enriched in Atox1 binding, these findings suggest that nuclear Atox1 functions as a transcription factor by direct binding to DNA, as well as by binding to DNA via other transcription factors (i.e., as a coactivator). In the context of in vitro studies using recombinant Atox1 protein and labeled DNA, Kahra et al. reported that Atox1 protein alone does not bind to DNA in vitro [24], which is in contrast to our findings [25]. Although the reason for this discrepancy is not clear, Atox1 may require other transcription factors or post-translational modification to stabilize its binding to DNA, as shown in transcription coactivator Yap and transcription factor TEAD [37]. However, the effect of cooperability of transcription factors on their affinity to DNA is complex, as shown in Oct4 and Sox2 [38,39]. Thus, it is conceivable that Atox1 binding with DNA directly or binding with other transcription factors as a cofactor may be context or agonist-dependent. For example, Atox1 binding with a specific motif or binding with another transcription factor may be determined by the physiological or pathological stimulants (inflammatory cytokines) in healthy or disease conditions such as diabetes and atherosclerosis. The molecular mechanisms underlying the transcriptional activity of Atox1 via Atox1-RE or other motifs clearly warrant further investigation.

Gene function annotation and pathway analyses revealed that signaling pathways enriched in Atox1-NLS-regulated gene targets are relevant not only to inflammation, such as TNFα signaling, but also to cell cycle, Hippo signaling, and p53 signaling pathways. We and others previously reported that Atox1 functions as a Cu-dependent transcription factor for cyclin D1 [25,27,31] or binding with an anaphase-promoting complex [40] to regulate cell proliferation. In the present study, we found that Atox1-NLS increased gene expression of cell cycle pathways and cell division, including cycle 25 A (CDC25A) [3] and CREBBP [41], which are known to induce cell proliferation, whereas it decreased CDC45, which is known to inhibit cell proliferation (Appendix A) [42]. These results suggest that nuclear-targeted Atox1 increases cell proliferation partially through these genes. Moreover, Atox1-NLS also increased inflammatory genes, TRAF1, and CSF1 but decreased anti-inflammatory gene MAP3K14 in ECs. It has been shown that Cu alters the conformation and transcriptional activity of the tumor suppressor protein p53 in human Hep G2 cells [26]. Consistent with this result, Atox1-NLS changed p53-regulated genes such as SESN2, MDM2, and GTSE1, which may reflect the Cu-dependent transcriptional activation of p53. Furthermore, Atox-NLS also regulates the Hippo signaling pathway (i.e., increase in TP73 gene expression and decrease in TEAD2 and FZD5 gene expression, see Appendix A). These results suggest that Atox1 may be important in regulating a transcriptional network directed by cell cycle, inflammatory, Hippo, and other signaling pathways. The functional significance of cell cycle and inflammatory gene regulation by nuclear Atox1 warrants further investigation.

The current study identified 248 genes directly regulated by nuclear Atox1 in which Atox1-RE (GAAAGA) is present within the +/− 3 kb region of the TSS. Using pathway analysis, we found that many genes regulated by Atox1 are involved in the inflammatory response and angiogenesis. Atox1 increased inflammatory genes (CREB3L3 [43], CD137 [44], CSF1 [45], IL5RA [46]) and decreased the anti-inflammatory gene (ADRB1 [47]). We previously reported that Atox1 plays an essential role in ROS-NFkB-dependent inflammatory neovascularization in response to hindlimb ischemia and proinflammatory cytokine TNFα by upregulating the NADPH oxidase organizer, p47phox, in ECs [16]. However, p47phox was not listed in Atox1-NLS-regulated genes, suggesting that Atox1-dependent p47phox gene upregulation may require additional factors or post-translational modifications, which are stimulated by TNFα. Paradoxically, overexpression of nuclear Atox1 increased antiangiogenic gene (SLC8A2 [48]) and decreased angiogenic gene (LMO2 [49], PIM1 [50]) expression. This suggests that persistent Atox1 localization in the nucleus may mimic pathological conditions, such as diabetes, in which reparative angiogenesis is impaired. It is also possible that too much Atox1 overexpression may cause irrelevant effects. In the case of atherosclerosis, the current study showed decreased atherosclerosis development on a high-fat diet in global Atox1-deficient mice, suggesting a proatherogenic role of Atox1. Addressing the role of endothelial nuclear Atox1 in atherosclerosis by using EC-specific Atox1^−/−^ mice, as well as EC-specific Atox1 overexpressing transgenic mice, is the subject of a future study.

RNA-Seq and ChIP-Seq analysis of Atox1-NLS overexpression, as well as a loss of function approach using siRNA, revealed CD137 as a new direct target of Atox1 to regulate both inflammatory response (monocyte adhesion) and ROS production. CD137 (TNFRSF9) is expressed in a variety of immunocytes and nonimmune cells, including ECs and smooth muscle cells. CD137 is involved in several known biological functions in ECs, including inflammation, angiogenesis, and apoptosis [51]. Previous studies have shown that the expression of CD137 is upregulated in atheroma-associated vascular cells in both mouse and human atherosclerotic plaque lesions in vivo [52,53]. In addition, we found that CSF1 and IL5RA are other direct targets of Atox1. CSF1 has been shown to be involved in atherosclerosis development via stimulating differentiation, growth, and survival of monocytes/macrophages [54,55,56,57,58,59,60]. Thus, the effect of CSF1 on the development of atherosclerosis is likely due to regulating monocyte/macrophage function. Previous studies showed that IL5RA is a hematopoietic receptor that plays a dominant role in eosinophil biology and can be influenced by proinflammatory cytokines, including IL-3, IL-5, IL-9, and GM-CSF [61,62,63]; however, the effect of IL5RA in regulating endothelial function has not been previously reported. Interestingly, we found that loss of CSF1 or IL5RA using siRNA inhibited Atox1-NLS-induced monocyte adhesion in ECs. Thus, Atox1-NLS-induced CSF1 and IL5RA may contribute to inflammatory responses in ECs. However, Atox1 siRNA significantly decreased CD137 and IL5RA mRNAs, but not CSF1 mRNA, in response to proinflammatory cytokines. These results suggest that proinflammatory cytokine-induced CD137 and IL5RA gene expression is Atox1-dependent and that CSF1 upregulation by cytokines requires additional transcription factors in ECs.

In conclusion, we demonstrated that Atox1-deficient mice showed decreased atherosclerosis development and identified downstream target genes regulated by nuclear Atox1 on the genome-wide scale in ECs. We showed that nuclear-targeted Atox1 overexpression in ECs increases inflammatory responses and ROS production via transcriptional-level activity in endothelial cells. Atox1 was shown to regulate multiple signaling pathways, including those involved in p53 signaling, Hippo signaling, and other signaling pathways. These findings provide new insights into nuclear Atox1 as a potential therapeutic target and a key transcriptional regulator of inflammatory gene expression in ECs.

## Figures and Tables

**Figure 1 cells-11-02919-f001:**
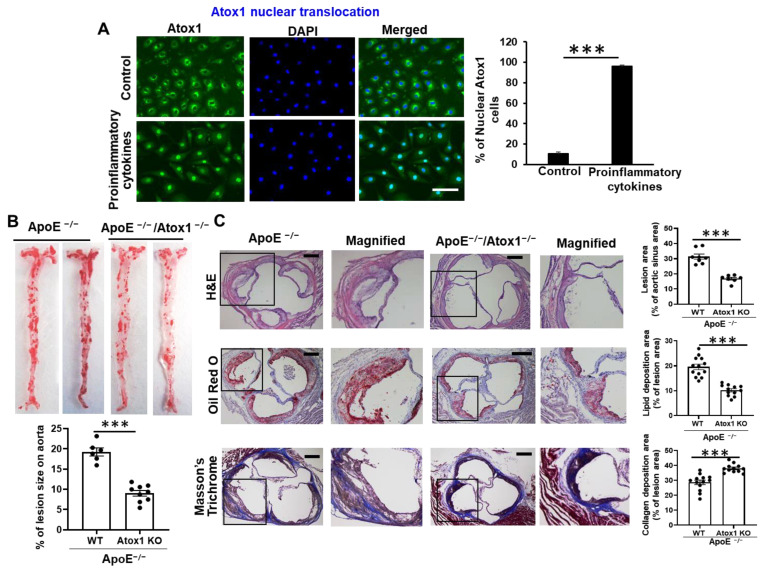
Role of Atox1 in inflammatory disease. (**A**). Immunofluorescence staining of Atox1. HUVECs stimulated with proinflammatory cytokine cocktail (TNF-α (10 ng/mL), IL-1β (10 ng/mL), and IL-6 (10 ng/mL) for 1 h were immunostained for Atox1 using anti-Atox1 Ab or nuclear marker DAPI. Percentage of Atox1+ cells in nucleus was shown in right (n = 3) Scale:50 μm. (**B**,**C**). Loss of Atox1 decreases atherosclerotic lesion in ApoE^−/−^ mice. ApoE^−/−^ or Atox1^−/−^ /ApoE^−/−^ were fed a Western diet for 16 weeks. (**B**). Representative images of Oil Red O stained-aortas (en face) from ApoE^−/−^ (n = 6) or Atox1^−/−^/ApoE^−/−^ (n = 9) mice after 16 weeks of Western diet. (Lower panel) Quantification data of lesion area. (**C**). Representative images and quantification data of cross-sections of aortic sinus of ApoE^−/−^ or Atox1^−/−^ /ApoE^−/−^ mice with 16 weeks of Western diet. Sections were stained with Oil Red O (lesion), hematoxylin and eosin (plaque), and Masson’s trichrome (collagen) with the magnified inset (black box) (n = 8–13). Scale bar: 200 μm. *** *p* < 0.001.

**Figure 2 cells-11-02919-f002:**
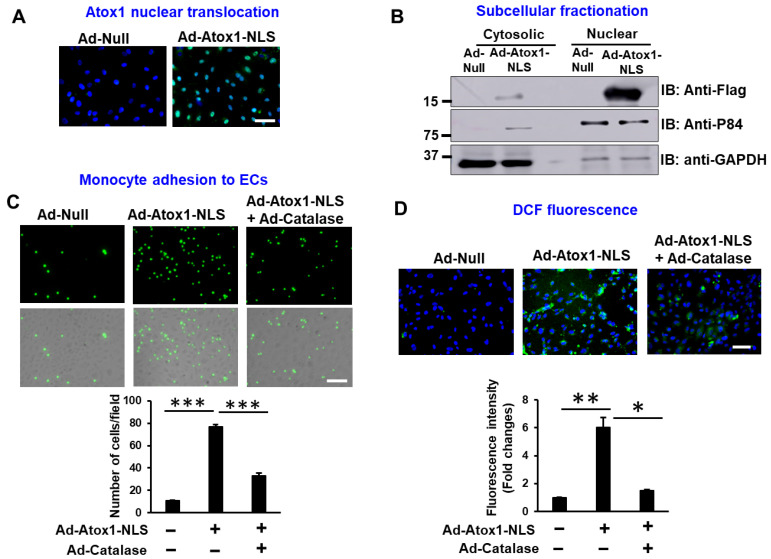
Nuclear Atox1 increases inflammatory response and ROS production in ECs. (**A**). Subcellular localization of nuclear-targeted Atox1 (Flag-Atox1-NLS) fusion proteins. HUVECs were transfected with adenovirus expressing Flag-Atox1-NLS (Ad-Atox1-NLS) or adenovirus alone (Ad-null), immunolabeled with antibodies for Flag tag (green) and nuclear marker, 4′,6-diamidino-2-phenylindole (DAPI) (blue). Scale: 50 μm. (**B**). Subcellular localization of Atox1 in Ad-Atox1-NLS or Ad-Null-transfected HUVECs. Nuclear or cytosolic fractions were subjected to Western blotting with anti-flag, anti-P84 (a marker for nuclear fraction), or anti-GAPDH (a marker for cytosolic fraction). (**C**). HUVECs were transfected with Ad-Catalase, Ad-Atox1-NLS, or Ad-Null for 48 h. The numbers of bound THP1 monocytes (fluorescently labeled) to ECs were measured with a fluorescence microscope. Right panel quantification (n = 5). Scale: 100 μm. (**D**). HUVECs were transfected with Ad-Atox1-NLS or Ad-Null in the presence or absence of Ad-Catalase for 48 h. Representative images for DCF fluorescence and DAPI staining (blue, nucleus marker) and quantification of fluorescence intensity (right, n = 3). Scale: 100 μm. *** *p* < 0.001, ** *p* < 0.01, * *p* < 0.05.

**Figure 3 cells-11-02919-f003:**
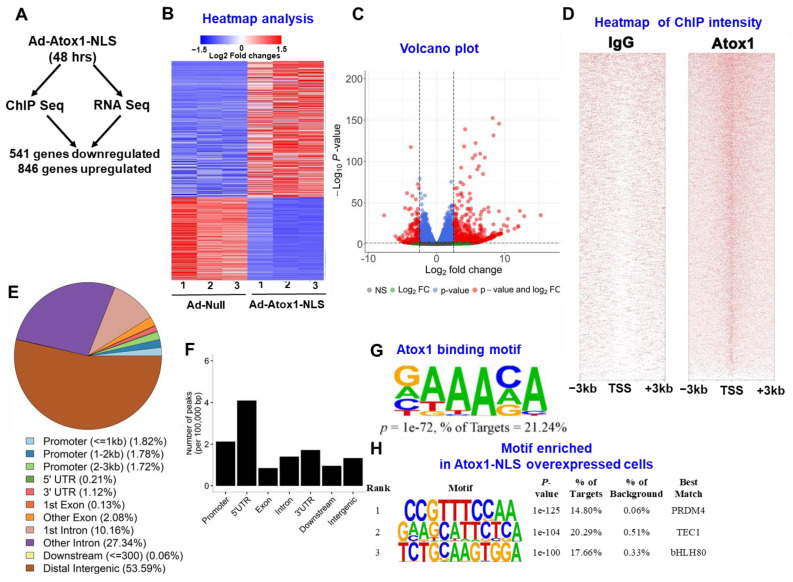
Genome-wide Atox1 gene targets identified by combining RNA-Seq and ChIP-Seq analysis. (**A**). Strategy for identifying Atox1 gene targets in HUVEC cells transfected with Ad-Atox1-NLS for 48 h. Atox1 gene targets represent genes that were bound by Atox1 and showed differential expression in response to Atox1-NLS. (**B**). Heat map of expression pattern for differentially expressed genes (DEGs) identified by combining ChIP-Seq and RNA-Seq. (**C**). Volcano plot of the pattern for all genes analyzed in RNA-Seq. Red dots represent DEGs with adjusted *p* < 0.05 and fold change >2.5 or <0.4. (**D**). Heatmap showing signal intensity of Atox1 peaks in ChIP-Seq in a 3 kb window centered around transcription start sites (TSSs) of genes across the genome. Atox1 was specifically precipitated from fragmented DNA-protein complexes by Monoclonal Anti-Atox1 antibody but not nonimmune IgG. Samples were used for ChIP-Seq. (**E**). Pie chart showing genomic distribution of Atox1-binding peaks. The genomic features (promoters, 5′ UTRs, exons, introns, 3′ UTRs, downstream, and intergenic regions) were defined based on RefSeq gene annotations. (**F**). Bar graph showing the distribution of Atox1-binding peaks in each genomic feature after normalization to the size of each category in 100 kb. (**G**). Atox1 binding motif (GAAAGA) in Atox1 ChIP-Seq peaks that correspond to the one identified in our previous work [25]. (**H**). Top 3 enriched motifs in Atox1 ChIP-Seq peaks.

**Figure 4 cells-11-02919-f004:**
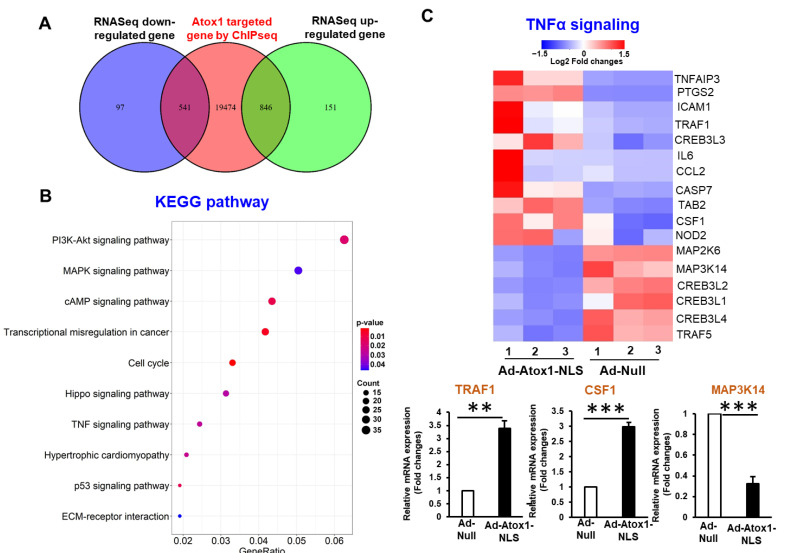
Analysis of nuclear Atox1-regulated genes. (**A**). Venn diagrams showing the overlap between differentially expressed genes (DEGs) identified by RNA-Seq and Atox1-bound genes identified by ChIP-Seq. (**B**). KEGG pathway enrichment analysis based on those 1387 overlapped DEGs in A. The top 10 enriched pathways ranked by the number of DEGs in the pathways are shown here. (**C**). Top: Heatmap showing the expression levels of DEGs belonging to the TNF signaling pathway in RNA-Seq. Color scale depicts range of log2 fold changes in gene expression. Bottom: Three selected DEGs in this pathway was validated by quantitative RT-PCR (n = 3) *** *p* < 0.001, ** *p* < 0.01.

**Figure 5 cells-11-02919-f005:**
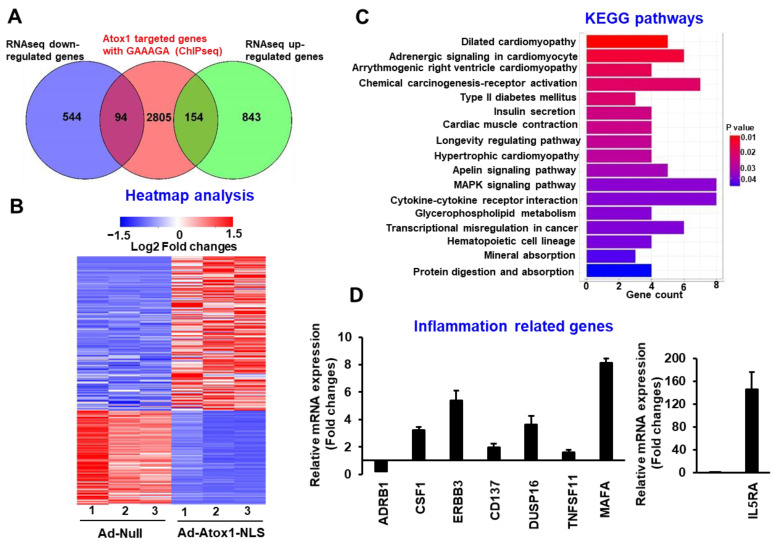
Genome-wide identification of direct Atox1 target genes. (**A**). Venn diagrams showing the overlap of DEGs directly regulated by Atox1, which are genes differentially expressed as identified in RNA-Seq and bound by Atox1 via Atox1 binding motif (GAAAGA) identified in ChIP-Seq. (**B**). Heatmap of expression pattern for those DEGs directly regulated by Atox1 as shown in (**A**). (**C**). KEGG pathway enrichment analysis of those 248 DEGs as shown in A. The 17 enriched pathways with *p* < 0.05 are shown here. (**D**). Subset of the Atox1 regulated gene targets was validated by quantitative RT-PCR (n = 3).

**Figure 6 cells-11-02919-f006:**
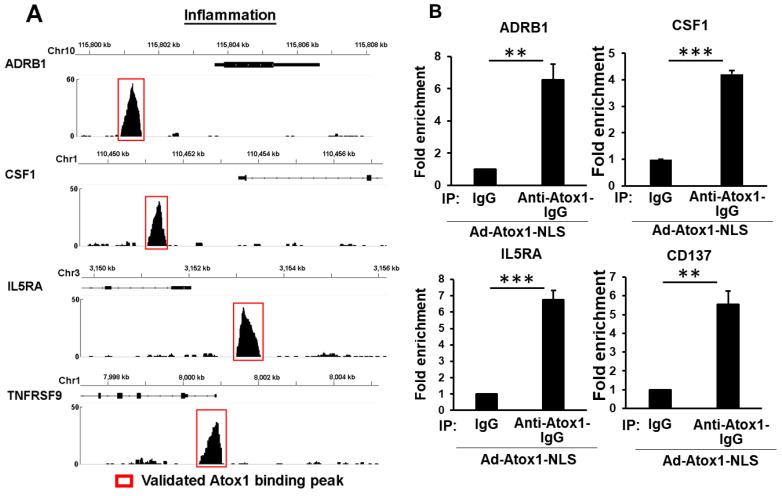
Validation of Atox1 binding peaks. (**A**). Representative tracks from ChIP-Seq results show the locations of Atox1 binding peaks proximal to the TSS of Atox1 targeted genes. Those peaks in red box were validated in (**B**). (**B**). HUVEC cells were transfected with Atox1-NLS for 48 h and subject to ChIP using monoclonal Anti-Atox1 Ab and mouse IgG. The enrichment of Atox1 binding peaks were determined by qPCR (n = 3). *** *p* < 0.001, ** *p* < 0.01.

**Figure 7 cells-11-02919-f007:**
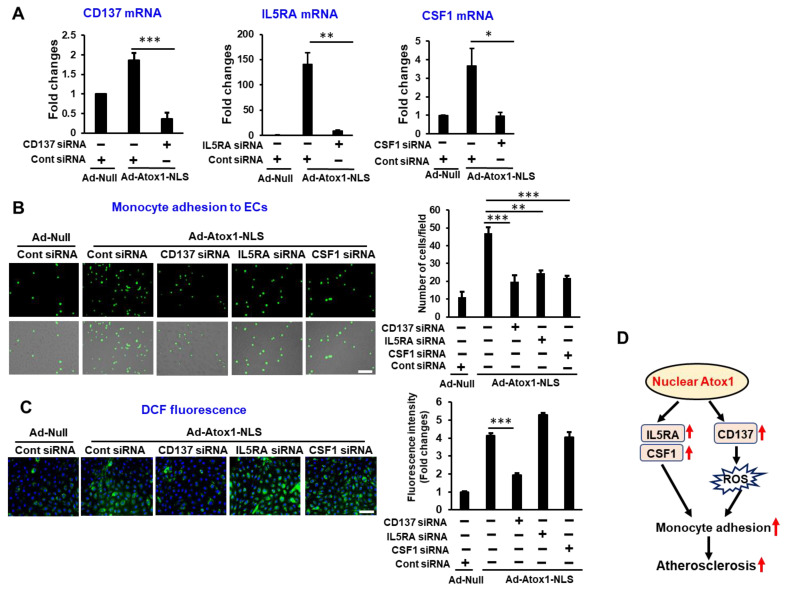
CD137, IL15RA, and CSF1 as new Atox1 targets to regulate inflammation and ROS production in ECs. ECs were transfected with CD137 or IL5RA or CSF1 siRNA. After 16 h, cells were transfected with Ad-Atox1-NLS or Ad-null for 48 h. (**A**). Cells were harvested and CD137 or IL5RA or CSF1 mRNA expression measured. Panel shows averaged data, expressed as fold change over Ad-Null (the ratio in Ad-Null-treated ECs transfected with Control siRNA was set to 1). (**B**). The numbers of bound THP1 monocytes (fluorescently labeled) to ECs were measured with a fluorescence microscope. Right panel quantification (n = 3). Scale: 100 μm. (**C**). Representative images for DCF fluorescence and DAPI staining (blue, nucleus marker) and quantification of fluorescence intensity (right) (n = 3). Scale: 100 μm. *** *p* < 0.001, ** *p* < 0.01, * *p* < 0.05. (**D**). Schematic diagram illustrating the mechanisms underlying the effect of Atox1-mediated inflammation and atherosclerosis.

## Data Availability

The data reported here are available from the corresponding author upon reasonable request.

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
