# Peer review of "Whole-Transcriptome Sequencing Analyses of Nuclear Antixoxidant-1 in Endothelial Cells: Role in Inflammation and Atherosclerosis"

_cells, 2022, doi:10.3390/cells11182919_

Round 1

Reviewer 1 Report

Endothelial cells (ECs) are important mediators in the development of atherosclerosis. The injured or inflamed ECs participate in recruiting inflammatory cells, which in turn exacerbates atherosclerosis. It has been reported that TNFα, a proinflammatory cytokine,  plays a critical role in promoting atherosclerotic plaque. Upon TNFα stimulation, a cytosolic copper chaperone Atox1 translocates into nucleus and promotes cell proliferation, and ROS production. Here, Sudhahar et al. explored the potential molecular mechanisms of Atox1 in endothelial cells upon TNFα stimulation using CHIP-seq and RNA-seq. They identified several potential downstream targets of Atox1. There are several issues needed to be addressed as follows:

(1)    Authors claimed that they focused on the potential mechanism of endothelial Atox1 in atherosclerosis. However, the global Atox1 knockout mice is not proper for this purpose.

(2)    When dissecting the potential signaling pathways, authors applied HUVEC as the cell model. However, it is quite different between venous and arterial ECs as studying atherosclerosis.

(3)    Authors hypothesized that the Atox1 nuclear translocation activated transcription to promote atherosclerosis and carried out a panel of experiments to identify the potential Atox1 targets. Thus, it is very important to show whether there is more Atox1 nuclear translocation in the ECs of atherosclerotic plaque. It is the basis of current study.

(4)    Authors used AD-Atox1-NLS to overexpress Atox1 and analyzed the CHIP-seq and RNA-seq. Besides of the cell type, there are several issues when using overexpression assay. First, authors haven’t confirmed that Atox1 overexpression may lead to atherosclerosis. Second, the overexpression efficiency is a potential issue whether too much Atox1 may cause irrelevant effects compared to the pathological process.

(5)    Authors should provide the reagent information, such as antibody for Atox1 CHIP-seq. Currently, Fig 2A : Authors use anti-FLAG Ab to show the nuclear location of Atox1 after overexpression. It is highly recommended to stain Atox1 as well. Moreover, western blots of total Atox1 is also needed to show the overexpression efficiency.

(6)    Z31112 is the GeneBank ID, instead of antibody information.

Reviewer 2 Report

In this study Varadarajan Sudhahar and collaborators found nuclear Atox1 downstream targets involved in inflammation and ROS production and provide insights into the nuclear Atox1 as a potential therapeutic target for the treatment of inflammatory diseases such as atherosclerosis.

The paper is quite innovative, and by using unbiased systemic approaches such as ChIP-Seq and RNA-Seq, authors identified several differently expressed genes regulated by nuclear Atox1. They found Atox1-NLS increases the genes involved in inflammation and ROS production in HUVECs, thus providing new insights into nuclear Atox1 as a potential therapeutic target for the treatment of inflammatory diseases such as atherosclerosis.

In my opinion, the data here obtained are of interest and makes the paper definitely worth to be published. Overall, the paper is interesting and well written. Only few issues should be addressed in order to improve this study for the readers of Cells.

1.      Figure 1A. Please make the figure and the legend clearer. It is not easy to read the control and the inflammatory cytokines treatment.

2.      The volcano plot in figure 3C is not readable. Also Figure 3D should be bigger and with a better resolution.

Minor points:

1.      Figure 3 B. Please better align the molecular weights

2.      Figure legends: in some legends the letters are not in bold

3.    The supplementary tables and figures are not always listed in the text in bold. Please be consistent

4.      In general figures should be better aligned. Figure 7B: please align the images

Reviewer 3 Report

The manuscript by Varadarajan Sudhahar and colleagues describes an important role of Atox1 in atherosclerotic lesions formation as well as differentially expressed genes upon overexpression of nuclearly localized Atox-1 in human endothelial cells. This was combined with CHIP-Seq to detect upregulated and downregulated factors with Atox1 binding motif sequence within 3kb of their promoters. The study is well designed, technically sound, clearly described and the results are not overinterpreted. Before accepting the manuscript, the authors should address following issues:

1. The authors could stimulate HUVECs with pro-inflammatory cytokines together with Atox-1 siRNA and analyse the expression of genes discovered as direct targets of Atox-1, e.g. CD137, IL5RA and CSF1  to evaluate whether these factors are regulated by Atox-1 in more physiological settings.

2. Stimulation with pro-inflammatory cytokines described in the Figure 1 caption are not mentioned in the Methods.

3. siRNA sequence for CD137 was provided however for IL5RA and CSF1 not

4. No information about Ad.catalyse in the Methods 

5. In the results: Of note, Ad.Atox1-NLS significantly increased monocyte adhesion (Fig.2C) and ROS production as measured by DCF fluorescence, which was inhibited by catalase overexpression (Fig.2D) in ECs. – latter reference to the figure should include Fig. 2C as well (.. overexpression (Fig.2C and D) in ECs)

6.      The fact that the ECs were stimulated with TNFa on Fig. 2C should be acknowledged in the Results 3.2 subchapter (not only in the Figure caption but also in the text

7.      The qPCR graphs represent expression relative to Ad.null group rather than 18S (y axis description)

8.      Primary Ab should be provided for immunofluorescent analysis

9.      Technical issue with Fig 1A labelling (the word control is on the picture)

10.   Abstract: “Inflammation, oxidative stress, and copper (Cu) play an important role in cardiovascular disease, including atherosclerosis, angiogenesis, and vascular remodeling” – angiogenesis and vascular remodeling are not cardiovascular diseases per se

11.   Introduction: “Antioxidant (Atox1) is known to function as… “ – it would be better to provide the full name of the protein here

12.   Results, 3.1 - … we used Atox1-/- mice crossed with either ApoE-/- mice (Atox1-/-/ApoE-/- ) mice or ApoE-/- (control) mice on a – would be better: we used either Atox1-/- mice crossed with ApoE-/- mice (Atox1-/-/ApoE-/- ) mice or ApoE-/- (control) mice

13.   Results, 3.1: “In contrast, Masson Trichrome staining of collagen in the aortic sinus was significantly (P<0.001) higher in the lesions from ApoE-/-/Atox1-/- mice as compared to control mice (Fig. 1B)” – should be Fig. 1C

14.   Several very minor spelling issues, e.g. 10ng/ml – should be 10 ng/ml; … such as the canakinumab Anti-inflammatory Thrombosis Outcomes Study CANTOS) – should be (CANTOS)

Round 2

Reviewer 1 Report

The authors have addressed my concerns either through new experiments or through discussion in the revised version.

Reviewer 3 Report

The authors responded to all issues raised by the Reviewer and imporved the manuscript.